# Mass action model of solution activity via speciation by solvation and ion pairing equilibria

Aaron D. Wilson [1✉], Hyeonseok Lee[1] & Caleb Stetson[1]

Solutes and their concentrations influence many natural and anthropogenic solution processes. Electrolyte and solution models are used to quantify and predict such behavior. Here we present a mechanistic solution model based on mass action equilibria. Solvation and ion pairing are used to model speciated solute and solvent concentrations such that they correlate to a solution's vapor pressure (solvent activity) according to Raoult's law from dilute conditions to saturation. This model introduces a hydration equilibrium constant ($K_{ha}$) that is used with either an ion dissociation constant ($K_{id}$) or a hydration modifier ($m$) with an experimentally determined ion dissociation constant, as adjustable parameters to fit vapor–liquid equilibrium data. The modeled solvation equilibria are accompanied by molecular dynamics (MD) studies that support a decline in the observed degree of solvation with increased concentration. MD calculations indicate this finding is a combination of a solvent that solvates multiple solutes, and changes in a solute's solvation sphere, with the dominant factor changing with concentration. This speciation-based solution model is lateral to established electrostatics-based electrolyte theories. With its basis in mass action, the model can directly relate experimental data to the modeled solute and solvent speciated concentrations and structures.

[1] Idaho National Laboratory, P.O. Box 1625 MS 2208, Idaho Falls, ID 83415-2208, USA. ✉email: aaron.wilson@inl.gov

As solution-based organisms who utilize and interact with a range of natural and anthropogenic solution systems, understanding solutions is critical for industry and foundational to understanding life itself. Solution speciation phenomena like solvation[1,2] and ion pairing[3,4] have been experimentally demonstrated[5] to contribute to various solution properties, thus attracting interest within emerging electrolyte theories[6]. Several research groups (ours[7], Zivitsas[8–11], Heyrokská[12–17], and others[18]) have taken the use of solution speciation a step further by directly using mass action equilibria (or processes) to model solution behavior without electrostatics. In these models the concentrations of solute species, generated by assembly processes (primarily solvation and ion pairing), are correlated to a solution's vapor pressure (solvent activity) via Raoult's law. Though these mass action models are a departure from established electrostatic-based electrolyte theories, they have the potential to explain molar behavior at concentrations generally considered non-ideal[19] as well as additional chemical phenomena, and does so while avoiding faulty assumptions such as the full dissociation of electrolytes. However, the mass action solution models proposed thus far address limited concentration ranges and lack a rigorous molecular basis. For example, in our previous work[7], solvation was quantified as a linear decline proportional to the increase in the solute concentration; while this method provided a mathematical fit to activity data, it was not based on known or expected reactions. There was no causal reason to relate the degree of solvation to the solute concentration. In this work, a mass action solvation equilibrium is proposed that can be readily correlated to vapor–liquid equilibrium (VLE) and other experimental data and is supported with molecular dynamics (MD) studies. This solvation equilibrium combined with ion-pair equilibria produces a robust, fully mechanistic solution model that models binary systems (solute and solvent) from dilute conditions to saturation.

## Results and discussion

**Solvation treatment in the mass action solution model.** In this work, the term *solvation* is adopted in lieu of coordination, as the energy of solvent molecules can be influenced by various physical phenomena in a statistically consistent way, for example, in (i) Lewis bases donating electron density to Lewis acids (e.g., classic coordination bond), (ii) hydrogen bonding (deterministic of the lower critical solution temperature behavior of amines and other materials[20,21]), and (iii) solute caging (e.g., formation of dissolved clathrate systems around solutes of differing polarity from the solvent[22]). Each solvent molecule present within the first solvation sphere of a solute is potentially *solvating*. Dividing bulk and solvating solvent into classes is consistent with the spatial response of solvent to even small amounts of charge[23]. For the purposes of this analysis, the distinction between solvating solvent and bulk solvent is defined by relative energies; simply, *solvation* is a stochastic process that induces a significant change to the chemical potential of a fraction of the total solvent.

The mechanism of solvation can be modeled through a series of sequential solvent dissociations (Eqs. 1–3) that can be merged (Eq. 4), simplified (Eqs. 5–8), and developed into an expression of speciated solute concentration (Eq. 9). These calculations use concentration of solute ($x_B$) species and solvent ($x_A$) species in mole fractions, as well as the concentration-dependent degree of hydration ($n$, $n_{(x_A)_{x_B}}$, and $n_{h_{x_B}}$), and a hydration modifier ($m$) to calculate composite solvation dissociation constants ($K_{hd}$) and its inverse, composite solvation association constants ($K_{ha}$). The association constant is then used to calculate a speciated solute concentration that takes hydration into account, Eq. 9.

$$(x_A)_n x_B \rightleftharpoons (x_A)_{n-1} x_B + x_A, \, K'_{hd} = \frac{[(x_A)_{n-1} x_B][x_A]}{[(x_A)_n x_B]} \quad (1)$$

$$(x_A)_{n-1} x_B \rightleftharpoons (x_A)_{n-2} x_B + x_A \quad , \quad K''_{hd} = \frac{[(x_A)_{n-2} x_B][x_A]}{[(x_A)_{n-1} x_B]} \quad (2)$$
$$\ldots \qquad\qquad\qquad\qquad , \quad \ldots$$

$$(x_A)_{n-m+1} x_B \rightleftharpoons (x_A)_{n-m} x_B + x_A, \, K'''_{hd} = \frac{[(x_A)_{n-m} x_B][x_A]}{[(x_A)_{n-m+1} x_B]} \quad (3)$$

$$K'_{hd} K''_{hd} \ldots K'''_{hd} = \frac{[(x_A)_{n-m} x_B][x_A]^m}{[(x_A)_n x_B]} \quad (4)$$

$$\frac{[(x_A)_n x_B]}{[(x_A)_{n-m} x_B]} K^*_{hd} = [x_A]^m \quad (5)$$

$$(n-m) \to n, \, \frac{[(H_2O)_n x_B]}{[(H_2O)_{n-m} x_B]} \to n(x_A)_{x_B} \quad (6)$$

$$n(x_A)_{x_B} K^*_{hd} = [x_A]^m \quad (7)$$

$$n_{h_{x_B}} = K_{ha}[x_A]^m \quad (8)$$

$$x_{B \pm ha} = \frac{x_B}{x_A - K_{ha}(x_A)^m x_B + x_B} \quad (9)$$

This solvation process, which is the primary model advancement over our previous work[7], is analogous to proton dissociation in a polyacid, where acid dissociation constants (pK_a) with closely aligned equilibrium energies affect the acid concentration-dependent pH of a solution (Eqs. 1–3). In such a scenario, individual pK_a can be reduced to a composite equilibrium in practical calculations (Eqs. 4 and 5). A frame of reference, based in the solvent's ($x_A$) concentration rather than solute's concentration ($x_B$) (Eqs. 1–5), is consistent with the proposition that solvation is more than coordination, and that it directly tracks the solvent's concentration. The distinction between solvent and solute framework is unnecessary, given that the two frameworks are equivalent under the conditions of interest when all solutes in solutions are fully equilibrated. The composite equilibrium, Eq. 5, features a ratio of solvent associated with a fully solvated solute [$(x_A)_n x_B$] relative to solvent associated with a depleted solute [$(x_A)_{n-m} x_B$]. As the depleted value, ($n-m$), approaches the fully solvated value, $n$, as expected in an equilibrated solution, the ratio [$(x_A)_n x_B$]/[$(x_A)_{n-m} x_B$] approaches the solute's degree of hydration [$n(x_A)_{x_B}$ or $n_{h_{x_B}}$], Equation 6. The equilibrium relationship can be rearranged to solve for the degree of hydration (Eqs. 7 and 8) which can be used to calculate a speciated solute concentration, $x_{B \pm ha}$, from the "absolute/anhydrous" mole fraction concentration, Eq. 9. This equilibrium expression designates hydration as exponentially proportional to the solvent concentration (Eqs. 4, 5, 7, and 8). The order of the solvent's exponential proportionality is determined by the energy of marginal dissociation events. A second order hydration term, $m$ (Eq. 8), is generally sufficient to model electrolytes from dilute conditions (0.1 molal) to saturation. For example, NaOH reaches a speciated mole fraction of ~0.9 within the model when combined with equilibrium ion pairing (Fig. 1). Thus, a high concentration is modeled for an electrolyte using two variables, indicating that a second order hydration modifier ($m$) is reasonable. The modeled concentrations in this work are higher than those achieved by Zavitsas' recent model, which reaches saturation for many systems but appears to be generally limited in prediction to ~7 molal, with a few exceptions[10].

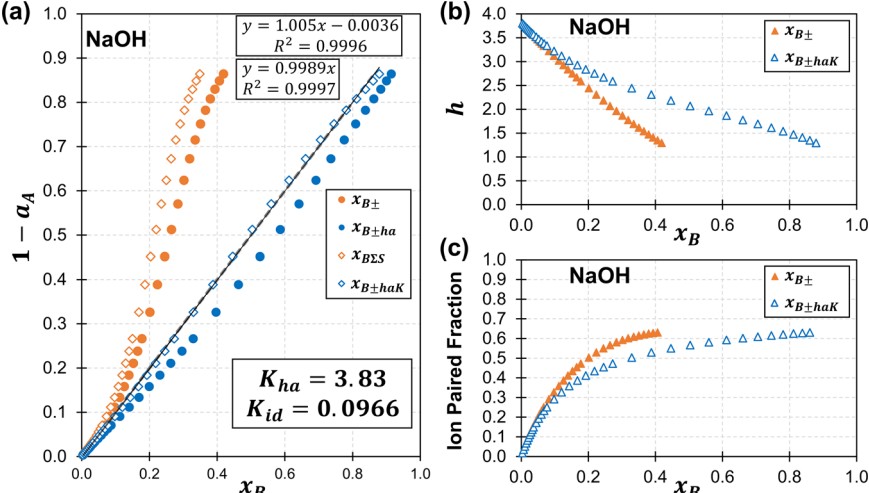

**Fig. 1 Comparison of fit and model components to the Raoult ideal for NaOH–H₂O activity data via speciation. a** VLE data[42,43] for NaOH modeled as ionized absolute/anhydrous salt ($x_{B\pm}$), ion paired ($x_{B\Sigma K}$, $K_{id} = 0.097$, Eq. 13), a second order hydration model ($x_{B\pm ha}$, $K_{ha} = 3.83$, Eq. 9) and ion paired with a second order hydration model ($x_{B\pm haK}$, $K_{id} = 0.097$, $K_{ha} = 3.83$, Eqs. 9 and 13). **b** Hydration and **c** fraction of ion pairing relative to concentration for the same systems.

**Ion pairing treatment in the mass action solution model**. The assumption that ions fully dissociate has evolved from a hypothetical starting point for modeling into a perceived physical reality. This misconception has been propagated by the theoretical structure of various activity coefficient models; however, experimental electrochemical, spectroscopic, and other methods indicate that all electrolytes exhibit some degree of ion pairing in solution[3,4]. Symmetric salts (e.g., 1-1 or 2-2 salts) can be addressed via a second order equilibrium as we previously described[7]. Asymmetric salts (e.g., 3-1 or 1-3 salts) are slightly more complex, but effective ion dissociation constant, $K_{id}$, can still be calculated relatively easily, Eqs. 10–14. The first dissociation of an anion in a 3-1 salt may be one of the few instances where full dissociation is a valid assumption; for example, aqueous AlCl₃ can be effectively modeled by assuming the full dissociation of one Cl⁻ (Eq. 10), the partial dissociation of the second Cl⁻ (Eq. 11), and no dissociation of the remaining Cl⁻ (Eq. 12). The intermediate dissociation event (Eq. 11) describes equilibrium speciation ($K_{id}$), which can be combined with the full dissociation event (Eq. 10) to yield the total number of solutes in solution (Eq. 13). Renormalization (Eq. 14) of these values provides the speciated mole ratio. This mass action solution model can be fit to experimental VLE data (as described in 'Methods' section 'Mass action solution model fitting mythology" and previous work[7]) such that the solution activity (vapor pressure) varies proportionally with the solvent's modeled mole fraction (i.e., the Raoult ideal) from dilute conditions to saturation. In the case of AlCl₃, a linear fit can be achieved with $K_{ha} = 7.08$ and $K_{id} = 0.027$. The treatment in Eqs. 10–14 is framed for more anions than cations, corresponding to 2-1 and 3-1 salts, but the process can be inverted for 1-2 and 1-3 salts or expanded for 2-3, 3-2, and other salts.

$$x_b^+(x_b^-)_n \rightleftharpoons x_b^+(x_b^-)_{n-m} + (n-m)x_b^-,$$
$$K_{id}' = \frac{[x_b^+(x_b^-)_{n-m}][x_b^-]^{n-m}}{[x_b^+(x_b^-)_n]} \quad (10)$$

$$x_b^+(x_b^-)_{n-m} \rightleftharpoons x_b^+(x_b^-)_{n-m-p} + (m-p)x_b^-,$$
$$K_{id} = \frac{[x_b^+(x_b^-)_{n-m-p}][x_b^-]^{m-p}}{[x_b^+(x_b^-)_{n-m}]} \quad (11)$$

$$x_b^+(x_b^-)_{n-m-p} \rightleftharpoons x_b^+(x_b^-)_{n-m-p-q} + (p-q)x_b^-,$$
$$K_{id}'' = \frac{[x_b^+(x_b^-)_{n-m-p-q}][x_b^-]^{p-q}}{[x_b^+(x_b^-)_{n-m-p}]} \quad (12)$$

$$x_{B\Sigma K'} = [x_b^-] + \sum [x_b^+(x_b^-)_{n*}] \quad (13)$$

$$x_{B\Sigma K} = \frac{x_{B\Sigma K'}}{x_{B\Sigma K'} + x_A} \quad (14)$$

**Application of the mass action solution model, periodic trends, and comparison to MD analysis**. This mass action model can also be applied to a neutral solute by ignoring ion pairing. Sucrose solution VLE data[24] can be modeled with a $K_{ha} = 3.82$ at 303.15 K. The hydration of neutral solutes matches a body of experimental work, including assessments made in Einstein's doctoral thesis[25] concerning solution viscosity.

Equilibrium hydration ($K_{ha}$, Eq. 9) combined with ion pairing ($K_{id}$, Eq. 14) effectively models all electrolyte solutions explored thus far (Table 2, Fig. 1, and Figs. S1–S36), apart from the lithium halides. When Eq. 9 is applied to lithium halides they model without ion pairing, conflicting with experimental data[4] despite full dissociation being an assumption in many electrolyte models. This overestimation of ion dissociation can be corrected by fixing the ion-pair dissociation constant, $K_{id}$, to literature values and allowing freedom in both the hydration modifier, $m$, and hydration equilibrium constant, $K_{ha}$, Table 2. The hydration modifier increases from 2.0 to ~2.4, consistent with a greater degree of hydration with closer energies of association. The higher level of hydration in lithium salts relative to other alkali metal salts is consistent with periodic trends in which cations with higher charge density experience a higher degree of hydration (Li⁺ > Na⁺ > K⁺ > Rb⁺ > Cs⁺), Table 2.

The ion pairing calculated for the other salts (where $m = 2$) is consistent with ion pairing values calculated by electrochemical means[4], Table 2. The $K_{id}$ values obtained from the electrochemical studies[4] and those derived here are not expected to be identical. The degree of hydration adjusts the "concentrations" resulting in an increased $K_{id}$ when calculated with Eq. 13 for the same amount of ion pairing per ion reported in the literature[4].

Thus, our $K_{id}$, while within the same order of magnitude as the electrochemically calculated ion pairing values, is consistently higher with increasing difference for more hydrated solutes.

The degree of hydration ($K_{ha}$) increases with the size of the halide for the $H^+$, $Li^+$, and $Na^+$ series. This trend could be attributed to a greater degree of dissociation; however, for $H^+$ and $Li^+$ halides there is an inconsequential change in dissociation as the anion changes. The most likely explanation is that larger halides have larger surfaces, requiring a greater degree of hydration via a caging mechanism, as shown by molecular dynamics (MD) simulations. MD was used to calculate the radial distribution function (RDF) for LiCl in $H_2O$ (Fig. 2a, b). MD calculations for both $Li^+$ and $Na^+$ halides show the fraction of solvent molecules within the first-solvation sphere increases with the halide volume ($I^- > Br^- > Cl^-$). MD calculations also indicate that $Cl^-$ requires a greater degree of solvation than $Li^+$. This is contrary to assumptions that cations are more solvated than anions due to a bias towards coordination bonds (which have a clear enthalpy) over the lower energy clathrate process involving the reorganization of existing bonds.

The first minimum in the RDF was set as a boundary and each solvent molecule within that distance was indexed as part of the first solvation sphere of one or more solutes. This calculation is distinct from the solute-based RDF calculation and allows the total fraction of solvent molecules involved in solvation to be calculated as alongside the solvation environment, Fig. 2e. The fraction of solvent molecules within the first solvation sphere calculated via MD (Fig. 2c) correlates well with the fraction of solvating solvent calculated with the analytical mass action solution model based on the values in Table 2. MD simulations employing classical force fields constitute the predictive modeling of dynamics that govern the intra-/inter-atomic parameters trained and developed with experimentally determined properties; individual simulations are often fitted against known thermophysical properties and phase equilibrium data (molecular model selections and simulation details are described in the 'Methods'). These two disparate methods/frameworks (statistical mechanics method and analytical mass action solution model), while developed from distinct datasets, yield similar concentration-dependent fractions of solvating solvent and bulk solvent.

The MD data also indicates a general decline in the degree of hydration with the increase of solute concentration, similar to the trend predicted by the equilibrium hydration model, but to a lesser extent. The MD study indicates that a significant portion of the decline in the degree of hydration results from joint solvation, Fig. 2. Equations 1–9 correlate solvent to a single solute, consistent with a coordination model but not necessary in a broader definition of solvation. Equations 1–9 thus imply that there are fewer solvent molecules in the vicinity of the solute as the solute concentration increases, "formally" ignoring the possibility that solvent molecules are shared between multiple solutes. However, joint solvation is captured in apparent/VLE fitted equilibrium constant ($K_{ha}$) because joint solvation (Eq. 15) is expressed in a manner that that matches dissociation (Eq. 1) and the related derivation (Eqs. 1–9).

$$2[(x_A)_n x_B] \rightleftharpoons [(x_A)_n x_B] \cdot [(x_A)_{n-1} x_B] + x_A,$$
$$K_{hd}^j = \frac{[(x_A)_{n-1} x_B][x_A]}{[(x_A)_n x_B]} \tag{15}$$

The RDF data for Li-O and Cl-H indicate that at 0.2 $x_{B\pm}$, 7.5% of the solvation attenuation results from changes in the coordination environment. However, at 0.4 $x_{B\pm}$, the influence of coordination environment on hydration attenuation has increased to 53.4%, Fig. 2a, b. This indicates that the degree to

which solvent sharing (Eq. 15) or solution environment changes (Eq. 1) affect solvation attenuation is concentration dependent. This influences how mixed solutions are considered in this mass action solution model. The apparent hydration of a solute will depend on its relative concentration, with dilute solutes appearing proportionally more solvated than more concentrated solutes.

A molecular perspective of a LiCl solution near saturation is graphically represented in Fig. 2d, illustrating the challenge of using implicit dielectrics to model concentrated solutions. The bulk dielectric of a solvent is meaningless when 90% of the solvent molecules are directly adjacent to one to four solutes, as is the case with saturated LiCl. Even for saturated NaCl, where ~70% of the solvent models as bulk solvent, there are only ~3 free solvent molecules for every solute and the concept of charge screening lengths becomes dubious. Direct interactions between solute and solvent, as utilized in a mass action-based solution model, is a reasonable basis for solution phenomena at saturation concentrations.

**Parameter space analysis of the mass action solution model.** The mass action-based fitting of VLE data consists of two opposing processes: ion pairing (Eqs. 10–14) that reduces the modeled solute concentration, and hydration (Eq. 8) that increases modeled solute concentration. The effects of these processes could become convoluted, resulting in multiple fits to experimental data (especially where experimental datasets are limited). NaCl is used to illustrate the fitting of the $K_{id}$ and $K_{ha}$ variable space in Fig. 3. NaCl was selected in part because it was challenging to fit with our previous model[7] that used a linear decline in hydration defined by the solute concentration.

With the introduction of equilibrium hydration based on the solvent concentration ($K_{ha}$, Eq. 8), NaCl data and other salts converge on specific combinations of $K_{ha}$ and $K_{id}$. As expected, there is overlap in the equilibrium effects that can be seen in the improved fit running from the high ion pairing and high hydration to low ion pairing and low hydration, top left to bottom right of Fig. 3. Despite this general trend, the differences between the two equilibrium functions result in clear maxima at $K_{ha} = 3.67$ and $K_{id} = 0.033$ for NaCl with an inverse residual sum square of $1.11 \times 10^5$. If the hydration modifier is allowed to vary in its degree, the inverse residual sum square increases to $6.89 \times 10^5$ with $m = 2.62$, $K_{ha} = 4.25$, and $K_{id} = 0.024$. Allowing three degrees of freedom to the hydration modifier provides a perfect fit between the modeled solvent concentration and solution solvent activity, with a slope of $1-a_A$ to $x_B$ of 1 and a $r^2$ value of one. However, there are potential fits with similar quality (i.e., poor convergence), indicating that three degrees of freedom for the hydration modifier results in overfitting. If merely three degrees of freedom results in overfitting, this mass action solution model presents an elegant match to existing VLE data. Future studies may suggest that the hydration value should be permitted freedom within a defined range, tied to other factors (e.g., $K_{ha}$), or fixed at another value; at present, it appears that fixing the hydration modifier at two ($m = 2$) is a reasonable working assumption, with the exception of the lithium halide series. That the lithium halide series requires a larger modifier to properly model ion pairing (discussed above) is a prime example of attention this modifier may require.

**Conclusion.** As development of an equilibrium expression of solvation depends on the concentration of solvent and solute, this work describes the first mass action solution model fully defined by chemical equilibria. It is capable of modeling solution activity from dilute to fully concentrated compositions for both electrolytes and non-electrolytes. The model is consistent with MD simulation,

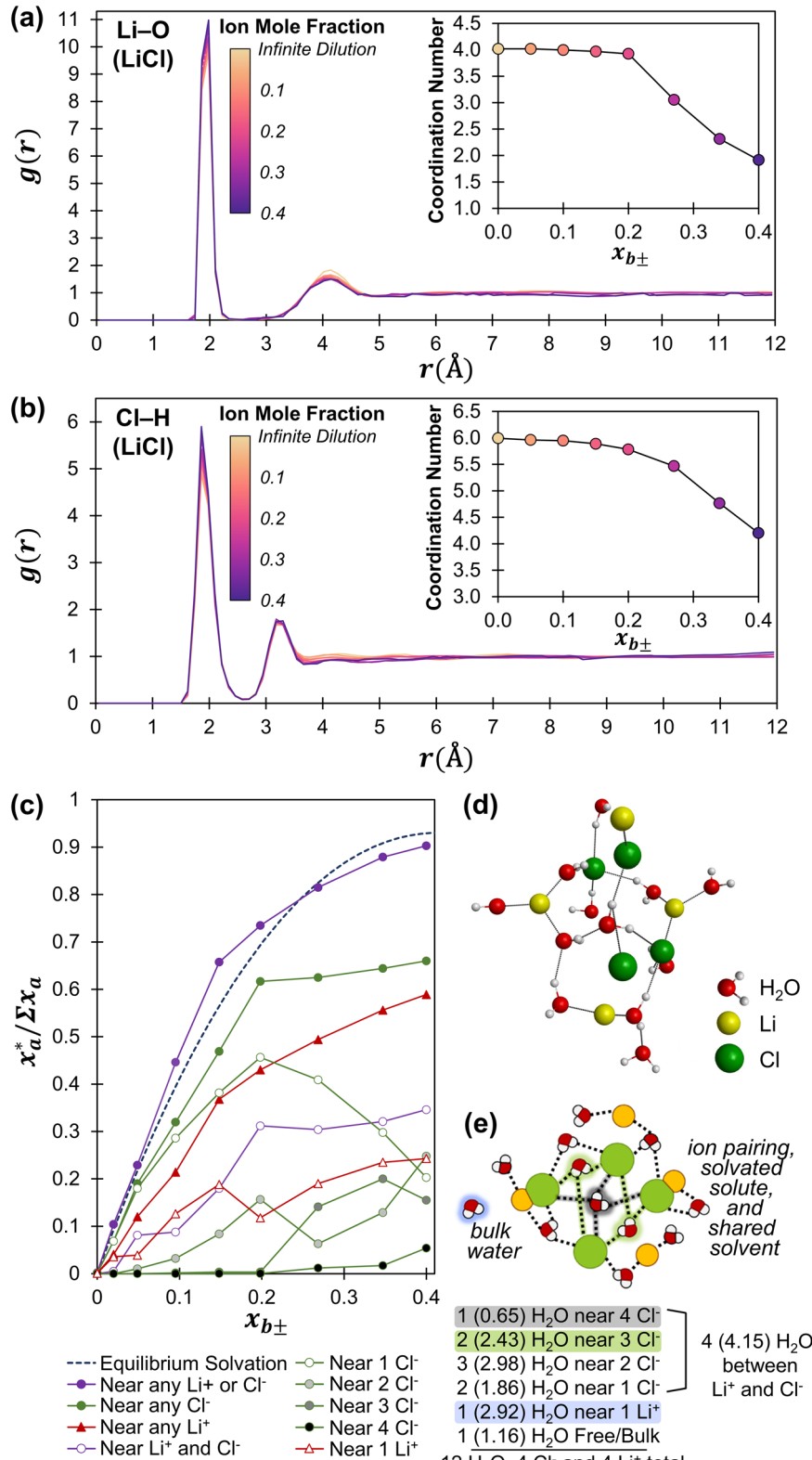

**Fig. 2 Molecular dynamics (MD) analysis of solute coordination environments and solvent sharing in the LiCl–H$_2$O system.** Calculated radial distribution functions (RDFs) for **a** Li–O and **b** Cl-H for LiCl as a solute in water for concentrations from 0.0 to 0.4 mole fraction, $x_{B\pm}$. Coordination number as a function of $x_{B\pm}$ is shown inset. **c** The near values are a fraction of the total solvent ($x_a*/\Sigma x_a$) defined by MD calculated RDF first-solvation sphere (minima/cutoff distances of Cl-H = 0.270 nm and Li-O = 0.258 nm). The number of solutes proximal to oxygen atoms are counted based on the distance at the minimum g(r) distance (Cl-O = 0.342 nm and Li-O = 0.258 nm). The equilibrium solvation total solvent fraction ($x_a*/\Sigma x_a$) is defined by ion-pair dissociation and hydration equilibrium constants ($x_{B\pm haK}$, $K_{id(fixed)}$ = 0.1, $K_{ha}$ = 4.78, $m$ = 2.41). **d** An equilibrated 3-D projection obtained in MD simulation of LiCl solvated by H$_2$O. **e** 2-D projection representing possible connectivity of atoms, rounded to whole integers (exact value in parenthetical) of the ionized 0.4 mole fraction system based on the MD calculations.

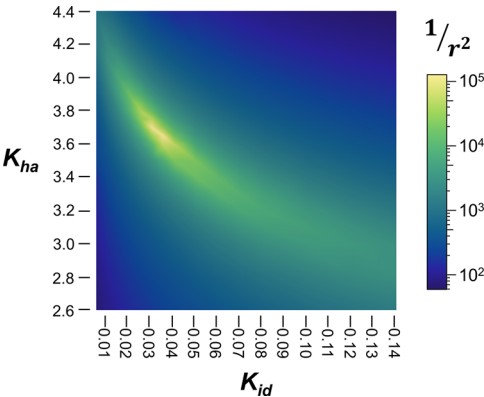

**Fig. 3 Convergence of the modeled variables in the NaCl–H$_2$O system.**
Variable space fit performance as defined by the inverse of the residual sum of squares (1/r$^2$) for fitting $K_{id}$ and $K_{ha}$ to experimental NaCl VLE data[42,43].

**Table 1 Parameters for the solutes and water models.**

| Molecules | Atoms/ions | charge | σ (Å) | ε (kcal/mol) |
|---|---|---|---|---|
| | Li+ | +1 | 1.616 | 0.1039884 |
| | Cl− | −1 | 5.520 | 0.0116615 |
| | Br− | −1 | 5.536 | 0.0303773 |
| | I− | −1 | 5.904 | 0.0417082 |
| Water[a] | O | −1.1128 | 3.1644 | 0.1852 |
| | H | 0.5564 | 0 | 0 |

[a]Potential parameters for flexible water model; $\theta_{eq}$ = 107.4 degree, $K_\theta$ = 43.9544 kcal/mol, Dr = 103.3893 kcal/mol, β = 2.2870 Å$^{-1}$, $r_{eq}$ = 0.9419 Å.

experimentally measured ion pairing (electrochemically and spectroscopic), and the magnitude of solvation values (determined from a range of experimental methods). The mass action solution model directly integrates with known and measurable processes rather than employing fitting parameters that minimize the difference between experimental data and a polynomial expansion of a theoretical model rooted in gas phase phenomena. This mass action solution model provides a pathway to integrate all measured solution properties, including vapor pressure, density, electrochemical behavior, characteristic spectra, diffusivity, heats of solvation, and solubilities into an interrelated theoretical framework. It is also a major departure from previous solution theory, bypassing an enormous body of work (Debye–Hückel, Pitzer, Davies, MSA, CPA, SAFT, NRTL, UNIQUAC). Mass action is conceptually lateral to electrostatics and thus not fundamentally inconsistent with these preceding theories; however, the contribution of mass action in the presented equilibria solution model, in contrast to established electrostatic theories, implies that both frameworks cannot be simultaneously valid. This mass action solution model's relationship to experimental values is compelling, given that electrostatic solution theories are focused primarily on "quality of fit" to specific datasets and lack means to differentiate between models. When system complexity increases (e.g., wider concentration ranges, mixed solutions, prediction of multiple properties, and acting on multiple scales) instrumental models tend to fail[26]. As this mass action solution model is tied to experimental data with parameters that correlate directly with measurable phenomena, it is suitable for continuous validation and improvement though a wide range of experimental measurements.

## Methods

**Molecular dynamics Intermolecular potentials.** queous lithium chloride solutions were previously investigated using MD simulations with 29 combinations of ion and water models by I. Pethes et al.[27]. This publication provided structural configurations for the ion and water system and the resulting Joung-Cheatham model of ion is in good agreement with experimental data. Hence, for the salt ions, the Joung-Cheatham (JC) set[28] were adopted since it has been previously validated for radial distribution function studies. Moreover, unlike Pethes's MD simulations with the rigid water molecule, we applied flexible water models, the TIP4P/2005f[29]. and SPC/Fw[30], which grants intra-molecular degrees of freedom to a rigid water model with increased effects on the local environment[31]. This occurs due to changes in dueto changes in molecular geometry affected by thermodynamic states that produce variations in the dipole moment. The total dipole moment changes as a results of variations in the relative orientation of the molecules[29]. The hydration number of Li$^+$ ions was directly calculated to validate models. Multiple ab initio MD studies calculated via different methods result in a hydration number of 4 for the Li$^+$ ion[32–35]. Our MD calculation with selected potentials closely matches previous results, with hydration number ranging from 4-4.6. The total interactions between molecules were calculated as the sum of Lennard-Jones (LJ) and Coulomb

interactions. The LJ function represents the Van der Waals forces, and describes the repulsive and attractive interactions between two molecules or atoms that temporarily create an induced dipole moment occurring by the motion of electrons. Likewise, the Coulomb function demonstrates particle interactions due to their permanent dipole moments that attract and repel particles from one another. The Lorentz−Berthelot (LB) mixing rule was employed in ion interaction. We applied four combinations of each solute ion (LiCl, LiBr, and LiI) and water models. For realistic simulation approach, we conducted further studies with the JC-TIP4P/2005f model combination (Table 1) based on its density profile comparison with experiment data[36].

**Molecular dynamics simulations.** MD simulations were performed in the isothermal-isobaric (NPT) ensemble with the LAMMPS simulation package[37] in a three-dimensional simulation box with periodic boundary conditions imposed in all directions. The initial non-overlapping liquid-like random molecular configurations were constructed by the software package PACKMOL[38]. The system was then allowed to equilibrate for a period of 5 ns with integration using a Nose–Hoover thermostat and barostat, where the density of the system converged to a mean value corresponding to the temperature and pressure conditions. Both the LJ and Coulomb interactions were modeled using a cut-off distance of 1.4 nm. The long-range coulombic interactions (beyond the 1.4 nm cutoff) were computed using the particle-mesh Ewald scheme (PME)[39] with an accuracy of 10$^{-4}$. The production simulations were performed for 20 ns, while the temperature and pressure were maintained constant with a coupled thermostat. The simulations in this work contained 3000 molecules of water as the solvent and 1 molecule of the solute to resemble infinite dilution conditions. And increased number of solute demonstrates that increasing molarity where solute is fully dissociated[40]. Monitoring of potential energy, pressure, and temperature during the production simulations confirmed that they stabilized with minor fluctuations, <1% for temperature. The molecular trajectories were sampled every 1000 steps to enable calculation of desired parameters.

As solute mole fraction increases, dissociation of the cation and anion ions in water was observed by MD simulations which calculated the radial distribution function (RDF), g(r)) and coordination numbers of ion-water as well as number of oxygen atoms (originating from molecules) near the ions. Our MD application method enables estimation of the fraction of water molecules involved in the dissociation of cation (Li$^+$) and anion (Cl$^-$, Br$^-$, I$^-$) in the system considering the position of water near the ions. The RDF measures the probability of a reference atom existing at the origin of a random frame and observation atom located in a spherical shell of thickness at a distance, r, from the reference atom. The average coordination (hydration) number is described by the number of observation atoms of present in a spherical shell of thickness dr, at a distance r from reference atom. The average coordination number can be calculated by integrating g(r) with respect to r[41]. Then, the average coordination numbers were calculated up to the first minimum of the g(r) curves from MD simulation. After the final MD trajectory information, we additionally worked for counting the number of oxygens based on the distance at the minimum of the g(r) curves. The number of water oxygens within a distance r (Li-O) from reference atom (Li ion) were counted, then the number of Cl ions within a distance r (Cl-O) from Li-O oxygens was obtained. This method provided the water solvent sharing fraction (simple positioning Li-O(water)-Cl). These results were validated by repeating the counting of water oxygens from Cl-O, a giving nearly identical results.

**Mass action solution model fitting mythology.** Datasets were obtained from R.A. Robinson, R.H. Stokes, Electrolyte Solutions: Second Revised Edition, Dover Publications, Incorporated, 2012, unless stated otherwise[42–49]. Datasets were fit using Excel's GRG Nonlinear Solver using the sum of squared deviation from the solvent activity residual $(1 − a_A)$ value. This parameter was then used to fit dissociation constant ($K_{ID}$) and hydration equilibrium ($K_{HA}$), or hydration modifier (m) and hydration equilibrium ($K_{HA}$) as specified in the text. Results are depicted in Table 2 and S1 as well as Figs. S1–S36. The dataset's solute absolute/anhydrous mole fraction ($x_B$) values were used to model various "modeled concentrations" as described by equations and text. Here the degree of hydration/solvation refers to a

**Table 2 Calculated $K_{ha}$ and $K_{id}$ using VLE Data at 298.15 K.**

| $K_{ha}$* $(K_{id})$ {lit $K_{id}$}† | H+ | Li+ | Na+ | K+ | Rb+ | Cs+ | NH4+ |
|---|---|---|---|---|---|---|---|
| NO3− | 2.15 (0.41) | 3.19 (0.35) | 1.34 (0.025) | 0.0# (0.0058) | 0.0 (0.0031) | 0.0# (0.0080) | 0.23 (0.022) |
| F− | | | 4.16# (0.012) | 3.53 (0.051) | 3.50# (0.045) | | |
| Cl− | 3.80 ($10^{6.3}$)** | $4.78x_A^{2.41}$ (0.10)** {0.041} | 3.67 (0.033) {0.011} | 2.89 (0.025) {0.012} | 2.73 (0.024) {0.014} | 2.45 (0.022) {0.018} | 2.37 (0.036) |
| Br− | 4.09 ($10^{8.7}$)** | $5.08x_A^{2.45}$ (0.12)** | 4.16 (0.033) | 2.85 (0.031) | 2.51 (0.024) | 2.50 (0.017) | 3.34# (0.022) |
| I− | 4.48# ($10^{9.4}$)** | $5.32x_A^{2.40}$# (0.15)** | 4.60# (0.039) {0.053} | 3.00 (0.037) {0.025} | 2.59 (0.023) | 2.06# (0.017) {0.014} | |
| OH− | | 2.53 (0.020) | 3.83 (0.097) | 3.82 (0.52) | | 3.39# (0.043) | |

Associated data plots can be found in the Supplementary Methods Figs. S1–S36.
*$K_{ha}x_A^2$ unless otherwise indicated. **Values manually fixed to order of magnitude to literature and periodic trend. #Data fitting limited to 3.5 molal and may be non-convergent. †{lit $K_{id}$} Dissociation constants obtained from electrochemical experiments[4].

non-integer believed to describe a stochastic association of solvent molecules with each solute (ion, ion pair, or molecule, as specified). This solvating solvent is removed from the bulk solvent used in calculating component speciated concentrations.

## Data availability

The data and calculations that support the findings of this study are available from the corresponding author upon request.

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

## Acknowledgements

This work was supported by the United States Department of Energy through contract DE-AC07-05ID14517. Funding was supplied by Idaho National Laboratory via the Laboratory Directed Research and Development Fund (LDRD) and National Alliance for Water Innovation (NAWI), funded by the US Department of Energy, Office of Energy Efficiency and Renewable Energy, Advanced Manufacturing Office, under Funding Opportunity Announcement DE-FOA-0001905.

## Author contributions

Aaron D. Wilson: Conceptualization, methodology, writing—original draft. Hyeonseok Lee: Formal analysis, writing—review & editing, visualization. Caleb Stetson: Writing—review & editing, visualization.

## Competing interests

The authors declare no competing interests
