## [Peer Review File · Communications Chemistry]

Reviewers' comments:

Reviewer #1 (Remarks to the Author):

In this manuscript, the authors have proposed the first mass-action solution model to use defined equilibria (solvation and ion pairing) to fully model solution activity for both electrolytes and non-electrolytes in the area of parameters from dilute to fully concentrated solutions. They proposed a mass-action solvation equilibrium which can be readily correlated to vapor liquid equilibrium and other experimental data and is supported with molecular dynamics studies. The authors obtained that this solvation equilibrium combined with ion-pair equilibria produces a robust, fully mechanistic solution model that models binary systems of solute and solvent, from dilute conditions to saturation. The authors illustrated the model on vapor-liquid equilibrium data for NaOH, composite solvation association constants (K_{ha}) and ion pairing (K_{id}) in solutions with the combinations of H^+ , Li^+ , Na^+ , K^+ , Cs^+ , Rb^+ , and NH_4^+ cations, and NO_3^- , F^- , Cl^- , Br^- , I^- , and OH^- anions, and then a speciated solute concentration that takes hydration into account. The authors noted the behavior of K_{ha} for the series of ions they described, contrary to the assumptions that cations are more solvated than anions due to a bias towards coordination bonds over the lower energy clathrate process involving the reorganization of existing bonds. The authors compared their findings to the molecular dynamics calculations. In conclusion, the authors claim that:

- (1) Their mass-action solution model provides a pathway to integrate all measured solution properties including vapor pressure, density, electrochemical behavior, characteristic spectra, diffusivity, heats of solvation, and solubilities into an interrelated theoretical framework;
- (2) It is a major departure from previous solution theory, bypassing an enormous body of work such as Debye-Hückel, Pitzer, Davies, MSA, CPA, SAFT, NRTL, UNIQUAC;
- (3) Mass-action is conceptually lateral to electrostatics and is not fundamentally inconsistent with these preceding theories;
- (4) The contribution of mass-action in the presented equilibria solution model that this framework cannot be simultaneously valid with the established electrostatic theories;
- (5) For wider concentration ranges and mixed solutions, predictions of multiple properties and acting on multiple scales using instrumental models tend to fail, whereas their mass-action solution model is tied to experimental data with parameters that correlate directly with measurable phenomena, given continuous validation against experiment.

This manuscript presents a considerable development of a new semi-empirical approach of a mass-action solution model which provides a path of fast calculation of vapor-liquid equilibrium data, solvation association constants, and ion pairing. This approach is interpolative, unlike statistical mechanics based molecular solvation theory which proved excellent prediction of solvation structure and thermodynamics of complex chemical and biomolecular systems. Even though, this manuscript deserves publication, as it introduces a fast interpolative approach to solvation thermodynamics.

Reviewer #2 (Remarks to the Author):

This article proposes a novel mass action equilibria-based approach to modelling the properties of electrolyte and non-electrolyte solutions. As the authors state this work bypasses a large body of work traditionally used for this problem. (Debye–Hückel etc.)

While I am certainly open to the idea of using alternative approaches to modelling electrolyte solutions. And the authors have outlined an intriguing approach to this, which could certainly be a promising alternative to the traditional approach. However, this article makes far too many vague and unsupported claims and far too much detail is obscured to make this article suitable for publication in anything like its current form in my view.

Firstly, some things that are too unclear.

Firstly, it is not clear how key properties are calculated. For instance, how is figure 1 a is calculated, does it include experimental data it is not clear why is NaOH data only given.

Something like a plot of all activity coefficients at 0.5 M with the model compared with experiment would be useful.

It is not clear how many fitted parameters are used for each electrolyte how does this compare to something like the Pitzer equations. How accurate is it compared with the Pitzer equations.

Some examples of the claims which are not clearly supported at all:

“however, the contribution of mass-action in the presented equilibria solution model, in contrast to established electrostatic theories, implies that both frameworks cannot be simultaneously valid.”

I don't know what justifies this claim.

“This mass-action solution model's relationship to experimental values is compelling, given that electrostatic solution theories suffer from significant underdetermination.”

Where is the clear demonstration that this model is less underdetermined than electrostatic models? In fact electrostatic models are very accurate at low concentrations where no parameters are required. So we know electrostatic models are 'correct' in the long range limit. The problem is they lack accurate short range interactions. It would be useful to provide an accurate description of the pros and cons of traditional models rather than just dismissing them with essentially no justification.

“prediction of multiple properties and acting on multiple scales) instrumental models tend to fail.”

Where are the examples of this?

“As this mass-action solution model is tied to experimental data with parameters that correlate

directly with measurable phenomena”

It is not clear to me that the parameters are ‘measurable’ at all if they were measurable, it should be possible to measure them and use the measurements to predict thermodynamic properties rather than relying on fitting. The notion that ions have fixed amounts of pairing or fixed number of solvating molecules is oversimplified. These things are in reality a constantly fluctuating and continuously relying on arbitrary cutoff chosen as the definitions. While this doesn’t mean this approach is impossible it is not important not to lose sight of the fact that this model can not actually reflect the underlying physical reality just as the idea of fully dissociating ions does not reflect the actual reality.

The comparison with MD is not particularly useful as it seems to just verify that fairly obvious properties are observed.

Reviewer #3 (Remarks to the Author):

We are offered a manuscript which describes a mass-action model for ionic solvation based on previous work by the authors (reference #6); however, unlike previous work, the model has an explicit basis in the specific chemical equilibria known to exist in the aqueous solutions of ionic compounds. The manuscript also presents us with MD simulations which confirm the physical basis of the model. The work is novel, the results impressive, and the quality of manuscript preparation excellent. The work makes a substantial advancement to the field of aqueous solvation, but should also be of interest to a more general audience. My opinion is that the manuscript is suitable for publication in Communications Chemistry. I additionally offer a few suggestions for minor revisions:

General Comments:

1. Throughout the manuscript, the authors could do better at articulating which aspects of the model are novel and which aspects are shared with, for example, reference #6. In particular, the novel component of the model is not even clearly stated in the abstract.
2. It would be nice to see at least some representative RDFs at different concentrations from the MD simulations, even in only in the SI.
3. In the introduction, it is claimed that previous mass-action models address limited concentration ranges. It would be nice to see a quantitative comparison of the model proposed here with previously proposed models.

Comments by line number:

171-174: The use of the word "basis set" is confusing here. Also, while it is generally true that force fields are fit to experimental data, it is not universally true.

225: Perhaps the authors mean "freedom" rather than "freed".

227: Perhaps "That the lithium" instead of "The lithium".

Reviewers' comments:

Reviewer #1 (Remarks to the Author):

In this manuscript, the authors have proposed the first mass-action solution model to use defined equilibria (solvation and ion pairing) to fully model solution activity for both electrolytes and non-electrolytes in the area of parameters from dilute to fully concentrated solutions. They proposed a mass-action solvation equilibrium which can be readily correlated to vapor liquid equilibrium and other experimental data and is supported with molecular dynamics studies. The authors obtained that this solvation equilibrium combined with ion-pair equilibria produces a robust, fully mechanistic solution model that models binary systems of solute and solvent, from dilute conditions to saturation. The authors illustrated the model on vapor-liquid equilibrium data for NaOH, composite solvation association constants (K_{ha}) and ion pairing (K_{id}) in solutions with the combinations of H^+ , Li^+ , Na^+ , K^+ , Cs^+ , Rb^+ , and NH_4^+ cations, and NO_3^- , F^- , Cl^- , Br^- , I^- , and OH^- anions, and then a speciated solute concentration that takes hydration into account. The authors noted the behavior of K_{ha} for the series of ions they described, contrary to the assumptions that cations are more solvated than anions due to a bias towards coordination bonds over the lower energy clathrate process involving the reorganization of existing bonds. The authors compared their findings to the molecular dynamics calculations. In conclusion, the authors claim that:

- (1) Their mass-action solution model provides a pathway to integrate all measured solution properties including vapor pressure, density, electrochemical behavior, characteristic spectra, diffusivity, heats of solvation, and solubilities into an interrelated theoretical framework;
- (2) It is a major departure from previous solution theory, bypassing an enormous body of work such as Debye-Hückel, Pitzer, Davies, MSA, CPA, SAFT, NRTL, UNIQUAC;
- (3) Mass-action is conceptually lateral to electrostatics and is not fundamentally inconsistent with these preceding theories;
- (4) The contribution of mass-action in the presented equilibria solution model that this framework cannot be simultaneously valid with the established electrostatic theories;
- (5) For wider concentration ranges and mixed solutions, predictions of multiple properties and acting on multiple scales using instrumental models tend to fail, whereas their mass-action solution model is tied to experimental data with parameters that correlate directly with measurable phenomena, given continuous validation against experiment.

This manuscript presents a considerable development of a new semi-empirical approach of a mass-action solution model which provides a path of fast calculation of vapor-liquid equilibrium data, solvation association constants, and ion pairing. This approach is interpolative, unlike

statistical mechanics based molecular solvation theory which proved excellent prediction of solvation structure and thermodynamics of complex chemical and biomolecular systems.

Even though, this manuscript deserves publication, as it introduces a fast interpolative approach to solvation thermodynamics.

We greatly appreciate the reviewer's thorough assessment of this manuscript.

Reviewer #2 (Remarks to the Author):

This article proposes a novel mass action equilibria-based approach to modelling the properties of electrolyte and non-electrolyte solutions. As the authors state this work bypasses a large body of work traditionally used for this problem. (Debye–Hückel etc.)

While I am certainly open to the idea of using alternative approaches to modelling electrolyte solutions. And the authors have outlined an intriguing approach to this, which could certainly be a promising alternative to the traditional approach. However, this article makes far too many vague and unsupported claims and far too much detail is obscured to make this article suitable for publication in anything like its current form in my view.

The authors would like to thank the reviewer for the valuable input. These comments will certainly improve the manuscript.

Firstly, some things that are too unclear.

Firstly, it is not clear how key properties are calculated. For instance, how is figure 1 a is calculated, does it include experimental data it is not clear why is NaOH data only given.

The manuscript originally stated, "This mass action solution model can be fit to experimental VLE data (as previously described [ref 6]) such that the solution activity (vapor pressure) varies proportionally with the solvent's modeled mole fraction (i.e. the Raoult ideal) from dilute conditions to saturation."

We have added a description of the method to the Supplemental Information. Text has been modified to read: "(as described in the SI and previous work [ref 6])"

The NaOH data presented in Figure 1 is provided as an example. The plots for all other modeled data can be found in the SI. Text has been modified to read: "Equilibrium hydration K_{ha} , Equation 9) combined with ion-pairing (K_{id} , Equation 14) effectively models all electrolyte solutions explored thus far (Table 1, Figure 1, and SI), apart from the lithium halides."

Something like a plot of all activity coefficients at 0.5 M with the model compared with experiment would be useful.

Thank you for the suggestion. This modeling approach intentionally does not generate activity coefficients, so a direct comparison to the activity is the best approach. Please refer to the extended datasets and plots included in the SI that feature experimental activity data and plots of model functions. It could also be possible to add fitting parameters to Table 1, but this may complicate the figure without adding value; all r^2 values are >0.99 .

To better indicate the changes made to the SI, we have added a sentence: "Equilibrium hydration (K_{ha} , Equation 9) combined with ion-pairing (K_{id} , Equation 14) effectively models all electrolyte solutions explored thus far (Table 1, Figure 1, and SI), apart from the lithium halides."

It is not clear how many fitted parameters are used for each electrolyte how does this compare to something like the Pitzer equations. How accurate is it compared with the Pitzer equations.

Thank you for the request for clarification. This model uses two parameters per fitting, out of three parameters that are possible to fit. To clarify this point, text has been added to the abstract, conclusion, and SI. The abstract text reads: "This model uses the hydration equilibrium (K_{ha}) and either an ion dissociation constant (K_{id}) or hydration modifier (m) as adjustable parameters to fit vapor liquid equilibrium data."

The main text includes this discussion: "Despite this general trend, the differences between the two equilibrium functions result in clear maxima at $K_{ha}=3.67$ and $K_{id}=0.033$ for NaCl with an inverse residual sum square of 1.11×10^5 . If the hydration modifier is allowed to vary in its degree, the inverse residual sum square increases to 6.89×10^5 with $m = 2.62$, $K_{ha}=4.25$, and $K_{id}=0.024$. Allowing three degrees of freedom to the hydration modifier provides a perfect fit between the modeled solvent concentration and solution solvent activity, with a slope of $1-a_A$ to x_B of 1 and a r^2 value of 1. However, there are potential fits with similar quality (i.e. poor convergence), indicating that three degrees of freedom for the hydration modifier results in overfitting."

The differences between these equilibrium constants and Pitzer parameters is substantial, and a *number of parameters* comparison (three versus two) is not valid. In future work, we look forward to completing a rigorous comparison between this model and those derived from Pitzer equations.

Some examples of the claims which are not clearly supported at all:

"however, the contribution of mass-action in the presented equilibria solution model, in contrast to established electrostatic theories, implies that both frameworks cannot be simultaneously valid." I don't know what justifies this claim.

Interestingly, this model addresses nearly all nonidealities without consideration given to electrostatic theories (as illustrated in Figure 1 and figures in the Supplementary Information). While the validities of the two distinct frameworks are not mutually exclusive, it is unlikely that they are simultaneously and/or equally correct.

"This mass-action solution model's relationship to experimental values is compelling, given that electrostatic solution theories suffer from significant underdetermination."

Where is the clear demonstration that this model is less underdetermined than electrostatic models? In fact electrostatic models are very accurate at low concentrations where no parameters are required. So we know electrostatic models are 'correct' in the long range limit. The problem is they lack accurate short range interactions. It would be useful to provide an accurate description of the pros and cons of traditional models rather than just dismissing them with essentially no justification.

The goal of this mass-action solution model is coherence with all available solution speciation data through model terms with a high degree of identifiability. Electrostatic theories use interaction and association parameters as fitting parameters, with quality of fit being the validity test; this makes electrostatic models difficult to falsify or "test". This difference is representative of the difference that exists between instrumental/engineering models and mechanistic models.

With this said the specific meaning of underdetermination (quality of fit versus differentiations between models) could be confused we are changing the text to: "This mass-action solution model's relationship to experimental values is compelling, given that electrostatic solution theories are focused primarily on *quality of fit* to specific datasets and lack means to differentiate between models."

"prediction of multiple properties and acting on multiple scales) instrumental models tend to fail." Where are the examples of this?

There are many examples of instrumental models that perform well in interpolation and poorly in extrapolation. A classic example would be fitting a curve with a higher degree polynomial without attention to its predictions outside the data fitting range. Higher polynomial fits generally perform better at interpolation and worse at extrapolation.

Rather than citing additional representative examples, it may be useful to refer to a recent review by Kontogeorgis that speaks specifically to this issue in electrostatic models. In this article, Kontogeorgis contends, "A common characteristic of these activity coefficient models for electrolytes is that they need a large number of adjustable parameters which are fitted to a wide range of experimental data. They are useful in engineering practice within the conditions (esp. temperature and concentration) used in their development, but they cannot be easily used for

extrapolations.” G.M. Kontogeorgis, B. Maribo-Mogensen, K. Thomsen, The Debye-Hückel theory and its importance in modeling electrolyte solutions, *Fluid Phase Equilibria*. 462 (2018) 130–152. <https://doi.org/10.1016/j.fluid.2018.01.004>.

“As this mass-action solution model is tied to experimental data with parameters that correlate directly with measurable phenomena”

It is not clear to me that the parameters are ‘measurable’ at all if they were measurable, it should be possible to measure them and use the measurements to predict thermodynamic properties rather than relying on fitting. The notion that ions have fixed amounts of pairing or fixed number of solvating molecules is oversimplified. These things are in reality a constantly fluctuating and continuous relying on arbitrary cutoff chosen as the definitions. While this doesn’t mean this approach is impossible it is not important not to lose sight of the fact that this model can not actually reflect the underlying physical reality just as the idea of fully dissociating ions does not reflect the actual reality.

As the reviewer has indicated, solute environments are in constant flux, and equilibria are expressions of constant flux. Ion pairing and solvation environments do not need to be defined by arbitrary cut offs. Ion pairs and larger clusters are evidenced by distinct spectroscopic signatures and other physical measurements, and impact physical properties of solutions. For example, pKa is an indispensable expression of ion pairing. The ion pairing modeled in this paper based on VLE data is consistent with ion pairing predictions based on electrochemical analyses. Solvation environments can also be measured in solution via x-ray and neutron diffraction RDF studies. This manuscript argues that including solvation and ion pairing may more closely reflect the underlying physical reality than assumptions of full dissociation of ions and no solvation. The objective of this work is to calculate equilibria and predict properties over a wider range than is possible to measure experimentally.

The comparison with MD is not particularly useful as it seems to just verify that fairly obvious properties are observed.

The changes in the effective degree of hydration could be a result of changes in an individual solute’s solvation environment and/or sharing of solvent. The molecular dynamics results indicate that sharing of solvent is the dominant factor over most of the concentration range, which is consistent with solution x-ray and neutron diffraction radial distribution function (RDF) studies. Understanding the interrelationship of these two solution phenomena is important in the application of this model to mixed salt solutions; we intend to study this topic in future work.

Reviewer #3 (Remarks to the Author):

We are offered a manuscript which describes a mass-action model for ionic solvation based on previous work by the authors (reference #6); however, unlike previous work, the model has an explicit basis in the specific chemical equilibria known to exist in the aqueous solutions of ionic compounds. The manuscript also presents us with MD simulations which confirm the physical basis of the model. The work is novel, the results impressive, and the quality of manuscript preparation excellent. The work makes a substantial advancement to the field of aqueous solvation, but should also be of interest to a more general audience. My opinion is that the manuscript is suitable for publication in Communications Chemistry. I additionally offer a few suggestions for minor revisions:

We greatly appreciate this reviewer's careful reading of the manuscript.

General Comments:

1. Throughout the manuscript, the authors could do better at articulating which aspects of the model are novel and which aspects are shared with, for example, reference #6. In particular, the novel component of the model is not even clearly stated in the abstract.

Thank you for this observation. The primary novelty of this work is the equilibrium hydration model (Equation 1-8). To clarify this departure from prior work, we have made changes to three locations in the text and the TOC figure.

Abstract (added): "This work introduces an equilibrium expression capable of modeling changes in effective hydration of solute, whether resulting from changes to the solvation sphere or to solute shared solvent."

Introduction (unchanged): "However, the mass-action solution models proposed thus far address limited concentration ranges and lack a rigorous molecular basis. For example, in our previous work,⁶ solvation was quantified as a linear decline proportional to the increase in the solute concentration; while this method provided a mathematical fit to activity data, it was not based on known or expected reactions. There was no causal reason to relate the degree of solvation to the solute concentration. In this work, a mass-action solvation equilibrium is proposed which can be readily correlated to vapor liquid equilibrium (VLE) and other experimental data and is supported with molecular dynamics (MD) studies. This solvation equilibrium combined with ion-pair equilibria produces a robust, fully mechanistic solution model that models binary systems (solute and solvent) from dilute conditions to saturation."

Just after Equation 1-9 (added): "This solvation process, which is the primarily model advancement over our previous work [ref 7], is analogous to proton dissociation in a polyacid,

where acid dissociation constants (pKa) with closely aligned equilibrium energies affect the acid concentration-dependent pH of a solution (Equations 1-3)."

Conclusion (added): As development of an equilibrium expression of hydration depends on the concentration of water and solute, this work describes the first mass-action solution model fully defined by chemical equilibria. It is capable of modeling solution activity from dilute to fully concentrated compositions for both electrolytes and non-electrolytes.

TOC: This now features a comparison of the solvation decline mechanisms as well as the quality of fit for the system using the primary equilibria.

2. It would be nice to see at least some representative RDFs at different concentrations from the MD simulations, even in only in the SI.

This is a very good suggestion. Molecular dynamics (MD) calculated radial distribution function (RDF) and coordination number for LiCl ions in water have been added to Figure 2. This calculation had not been done for the full concentration range. It indicated that the degree to which solvation attenuation with concentration is driven by changes in solvation environments versus solute molecules sharing of solvent. It has resulted in changes after Equation 15 and the abstract. This will also impact our future work.

3. In the introduction, it is claimed that previous mass-action models address limited concentration ranges. It would be nice to see a quantitative comparison of the model proposed here with previously proposed models.

Zavitsas' original model that utilized a single hydration number was limited to eutectic point concentrations. His recent model explores more systems and reaches higher concentrations, but the concentration limit is highly variable. Heyrokská models don't indicate a clear concentration range for validity but appear to reach 6 molal in some cases. Our previous model (reference 6) went to saturation but achieved a lower quality fit than the model presented in this work.

We added/modified the statement "Thus, a high concentration is modeled for an electrolyte using two variables, indicating that a second order hydration modifier (m) is reasonable. The modeled concentrations in this work are higher than achieved by Zavitsas' recent model, which reaches saturation for many systems but appears to be generally limited to prediction up to ~7 molal, with a few exceptions.¹⁰"

Comments by line number:

171-174: The use of the word "basis set" is confusing here. Also, while it is generally true that force fields are fit to experimental data, it is not universally true.

Thank you for bringing this to our attention. We revised the following sentence to the main text in the eighth paragraph in the section 2.0 Result and Discussion:

"MD simulations employing classical force fields constitute the predictive modeling of dynamics underlying intra-/inter-atomic parameters that have been trained/developed using experimental properties, with individual runs often fitted against known thermophysical properties and phase equilibrium data (molecular model selections and simulation details are in SI)."

225: Perhaps the authors mean "freedom" rather than "freed".

Thank you, this has been corrected in the manuscript.

227: Perhaps "That the lithium" instead of "The lithium".

Thank you for pointing this out. The sentence has been revised to read: "**That the lithium** halide series requires a larger modifier to properly model ion-pairing (discussed above) is a prime example of attention this modifier may require."

REVIEWERS' COMMENTS:

Reviewer #1 (Remarks to the Author):

In the revised version of this manuscript, the authors adequately addressed the questions raised by the Reviewers. The revised manuscript deserves publication in the present form.

Reviewer #2 (Remarks to the Author):

I am satisfied with the changes made by the authors and am happy to recommend acceptance

Reviewer #3 (Remarks to the Author):

The authors have produced a thoughtful response to the referees and made significant improvements to the manuscript. I recommend publication.